# Effectiveness of the Organic Acid-Based Antimicrobial Agent to Prevent Bacterial Contamination in Fish Meal

**DOI:** 10.3390/ani12233367

**Published:** 2022-11-30

**Authors:** Wattana Pelyuntha, Ananya Yafa, Baramee Charoenwong, Kitiya Vongkamjan

**Affiliations:** 1Department of Biotechnology, Faculty of Agro-Industry, Kasetsart Univeristy, Bangkok 10900, Thailand; 2LinQ Technology Corporation, 29/7 Thakai, Chachoengsao 24000, Thailand

**Keywords:** antimicrobial agent, feed additive, fish meal, microbial contamination, organic acids, *Salmonella*

## Abstract

**Simple Summary:**

Raw ingredients for farm animal feed production are highly contaminated with spoilage microorganisms and foodborne pathogens that affect the quality and sensory aspect, especially fish meal, a major feed ingredient. The contamination of pathogens in fish meal for feed production is remain as an important safety issue where pathogens may spread into food animals and later causes illness in humans when food animals are consumed. Therefore, chemical treatments are usually applied for extending the quality and shelf-life of fish meal. Organic acids are an alternative chemical treatment instead of formaldehyde-based products that can reduce the existing contamination and prevent recontamination of *Salmonella* in feed supply chain. In present study, SALTEC 514^TM^, an organic acid-based antimicrobial agent, was applied in fish meal during storage for preventing the risk of microbial contamination and recontamination in fish meal. Our findings here will provide an effective alternative for control of microbial contamination in the feed production industry thereby improving feed and food safety.

**Abstract:**

Animal feed production is an important step of the food animal production chain in a farm-to-table model. The contamination of raw ingredients with foodborne pathogens in feed production remains as an important safety issue where pathogens may spread into food animals to cause illnesses in humans when affected food animals are consumed. In the present study, we aimed to examine the quality and microbial contamination of fish meal and to investigate the effectiveness of the organic acid-based antimicrobial agent SALTEC 514^TM^ against *Salmonella* to prevent bacterial contamination in fish meal. Fish meal samples (*n* = 4) collected from feed mills at different locations were analyzed for protein and total volatile basic nitrogen (TVBN) content to assess their nutritional value and freshness, and its microbiological quality. The protein and TVBN content ranged from 53.2 ± 3.1 to 67.5 ± 2.3 g/100 g and 73.8 ± 4.5 to 100.4 ± 11.2 mg/100 g meal, respectively. Total plate count of the fish meal samples ranged from 2.0 ± 0.3 to 4.5 ± 0.5 log units, whereas suspected foodborne bacteria, *Escherichia coli* and *Salmonella*, were not detected in all samples. Fish meal samples were artificially contaminated (day 0) and re-challenged (day 30 and 90) with *Salmonella* Enteritidis (3 log CFU/g) to test for the effectiveness of SALTEC 514^TM^, an organic acid-based antimicrobial formulation, in preventing *Salmonella* contamination and recontamination during storage. SALTEC 514^TM^, when applied at three different doses, was found to reduce the number of *Salmonella* in monitored samples after one day of storage. A low dose of 0.5 kg/ton SALTEC 514^TM^ prevented *Salmonella* recontamination from occurring in fish meal samples stored for 37 days. In medium (1.0 kg/ton) and high doses (3.0 kg/ton), applications of SALTEC 514^TM^ prevented the *Salmonella* recontamination for a maximum storage duration of 97 days. The application of SALTEC 514^TM^ in fish meal and/or other feed ingredients may prove to be a safe alternative to reduce the microbial load, especially of foodborne-related microorganisms, to contribute to feed and food safety.

## 1. Introduction

Raw ingredients for farm animal feed production such as fish meal may originate from a variety of locations [1]. If the ingredient was exposed to potential sources of *Salmonella*, it can become contaminated to pose a serious concern for the feed industry. Previous studies have reported the occurrence of *Salmonella* spp. in feed ingredients and feed as a major risk of *Salmonella* contamination in feed mills [2,3]. Contamination by *Salmonella* and other undesirable microorganisms can occur at various stages of production, shipping, processing, or storage, potentially resulting in contamination of finished feed [4,5].

Fish meal is produced from fish meat and residues or known as a by-product of fish oil production. It is mainly used as an animal feed ingredient, especially for poultry, cultured fish and shrimps, and livestock feed [1]. Fish meal prepared from fish such as mackerel, sardine, and anchovy is a main source of histamine in feed [6]. Fish soluble is a by-product of canned tuna processing produced from tuna-pressed liquid. It is then processed into fish soluble concentrate (FSC) which is high in protein and amino acid content, making it useful as an ingredient for aquaculture and animal feed [7].

*Salmonella* is able to persist for many years in dry environments such as those found in feed mills, grain stores and feed bins. Once resident, *Salmonella* can be difficult to eradicate [7]. Resident strains can enter feed processing equipment, including after critical control steps such as heat treatment, and may multiply in situ. This may lead to intermittent or continuous contamination of compound feed during the milling process [7]. *Salmonella* can also survive in the environment of farms and fish processing. If wildlife or rodents have access to feeding systems once feed has been delivered, there is potential for new contamination of feed at that stage [7,8].

Formaldehyde-based products were previously used in the European Union (EU) to counteract feed contamination under Directive 98/8/EC. A change in legislation resulting in Regulation (EU) No. 528/2012 then required formaldehyde to be approved as a feed additive under Regulation (EC) No. 1831/2003. In December 2017, this approval was denied by the EU Commission’s Standing Committee on Plants, Animals, Food and Feed (SCoPAFF). Thus, the use of organic acids has become an alternative approach to prevent feed contamination. Organic acid treatments can reduce existing contamination and prevent recontamination further along the supply chain. Organic acids are generally recognized as safe (GRAS) for use as direct feed and food additives (i.e., as chemical preservatives in animal feeds and as antimicrobial/flavoring agents in human foods) [9]. The present study investigated the use of SALTEC 514^TM^, a liquid antimicrobial agent containing formic acid and propionic acid (85% total acids) with surfactant, to reduce the risk of microbial contamination in feed ingredients where fish meal samples were experimentally contaminated with *Salmonella* at start and after a storage period for 97 days.

The objective of this study was to investigate the effectiveness of SALTEC 514^TM^ against *Salmonella* contamination and recontamination in fish meal during storage. The application of SALTEC 514^TM^ may provide an effective alternative for control of *Salmonella* in the feed production industry, thereby improving feed and food safety.

## 2. Materials and Methods

### 2.1. Fish Meal Samples and Antimicrobial Agent

Four fish meal samples, assigned as SM, SO, K, and T were obtained from four feed mills in May 2021. Samples were kept in sealed bags and stored at room temperature for further experiments. SALTEC 514^TM^, the organic acid-based antimicrobial agent, was obtained from the LinQ Technology Corporation, Chachoengsao, Thailand.

### 2.2. Protein and Total Volatile Basic Nitrogen Contents

To determine protein and total volatile basic nitrogen (TVBN) contents, the collected fish meal samples were analyzed by a commercial laboratory (S.P.T.N. Logistics Co. Ltd., Samut Sakhon, Thailand) using an in-house protocol. Protein content was determined by the Kjeldahl method using cupric and potassium sulfate as catalyst. Total volatile basic nitrogen content was determined by distilling the volatile bases from a weighed sample of meal (3 g) to which magnesium oxide was added into 30 mL of saturated boric acid. The solution was further titrated with 0.02 M hydrochloric acid using methyl red as an indicator.

The TVBN values were calculated and expressed as mg N per 100 g meal on a dry basis. The protein and TVBN contents of fish meal ranged from 53.2 to 67.5 g/100 g and 73.8 to 111.0 mg N/100 g meal, respectively (Table 1). The highest protein content was observed in the fish meal sample obtained from feed mill T, while the lowest was from feed mill SM. The highest TVBN content was observed in the sample from feed mill as 100.4 ± 11.2 mg/100 g meal, followed by feed mill T and SM. The lowest TVBN content was found in a sample from feed mill SO as 73.8 ± 4.5 mg/100 g meal.

### 2.3. Total Plate Count and Detection of Escherichia coli

Fish meal samples were enumerated for total bacterial count following Association of Official Agricultural Chemists (AOAC) method. Fish meal sample (25 g) was serially 10-fold diluted using a phosphate-buffered solution (PBS). An aliquot of 0.1 mL was plated onto PCA plates. Colonies were counted after incubation at 37 °C for 24 ± 2 h. To investigate for the presence of *E. coli*, an alternative method of ISO16140-2:2016 was followed, where an aliquot of 0.1 mL was plated onto ChromID^®^ Coli (COLI ID-F, bioMérieux, Marcy l’Étoile, France) and incubated at 44 °C for 24 ± 2 h. Pink colonies of *E. coli* that appeared in the agar plate were further observed and enumerated.

### 2.4. Detection of Salmonella spp.

Fish meal samples were investigated for the presence of *Salmonella* spp. following the standard protocol of ISO 6579:2017. Approximately 25 g of fish meal was enriched with 225 mL of buffered peptone water (BPW) at 37 °C for 18 ± 2 h. A mixture of 0.1 mL was transferred into 10 mL Rappaport-Vassiliadis soya peptone broth (RVS) broth (#CM0866) and incubated at 41.5 °C for 24 ± 3 h. The same mixture of 1 mL was transferred into 10 mL of Muller-Kauffmann Tetrathionate-Novobiocin (MKTT-n) broth (#2910-13-1048T) and incubated at 37 °C for 24 ± 3 h. After incubation, a loopful of culture medium was streaked on a xylose lysine deoxycholate (XLD) agar plate and Hektoen Enteric (HE) agar plate (#CM0419) and incubated at 37 °C for 24 ± 3 h. Suspected colonies of *Salmonella* were re-streaked on TSA plate and confirmed with preliminary biochemical tests including lysine iron agar (LIA) (#CM0381), triple sugar iron (TSI) agar (#CM0277), and urea agar (#CM0053). Then, suspected colonies of *Salmonella* were further selected for serotyping by an agglutination latex test carried out by a commercial laboratory (S & A. Reagents Lab. Co., Ltd., Part., Bangkok, Thailand).

### 2.5. Treatment of Fish Meal Using SALTEC 514^TM^ against Salmonella

SALTEC 514^TM^ (LinQ Technology Corporation, Chachoengsao, Thailand) was applied to uninoculated fish meal samples from four feed mills (SM, SO, T, and K) at low, medium, and high concentrations, representing 0.5 kg/ton, 1 kg/ton, and 3 kg/ton, respectively. Each fish meal sample was kept in sterile plastic bags for 2 h at 25 °C prior to monitoring and conducting SALTEC 514^TM^ effectiveness tests. A positive control sample with SALTEC 514^TM^ omitted from each feed mill was included.

The overnight culture of *Salmonella enterica* subsp. *enterica* serovar Enteritidis S5-371 at a final concentration of 10^3^ CFU/g was added into each fish meal sample, and samples were kept at 25 °C. The count of *Salmonella* in each sample was then determined by direct count method on XLD agar plate after storage for 1 and 7 days. On day 30 and 90, the fish meal samples stored at 25 °C were again subjected to 1 mL of the overnight *Salmonella* culture in phosphate-buffered solution (PBS) at the same final concentration as a recontamination of *Salmonella* in the samples. The *Salmonella* count was enumerated on days 31, 37, 91, and 97, respectively. All treatments were run in triplicates. For qualitative study, the presence and absence of *Salmonella* in the samples was also confirmed by the *Salmonella* standard protocol of ISO 6579: 2017 as previously described.

### 2.6. Statistical Analysis

Statistical analysis was performed using SPSS Statistics software, version 22.0 for Windows (IBM Corp, Armonk, NY, USA). Data of *Salmonella* count at each storage time and each dosage were subjected to analysis of variance followed by Tukey’s range test. A difference was considered statistically significant if *p* < 0.05.

## 3. Results

### 3.1. Total Plate Count and Occurrence of E. coli and Salmonella in Fish Meal

Fish meal samples obtained from the two of the four feed mills (Table 2) showed low total plate counts as 2.0 ± 0.3 log CFU/g for feed mill SO and 2.3 ± 0.2 log CFU/g for feed mill SM. Higher counts were observed among samples from two other plants: 4.2 ± 0.4 log CFU/g for feed mill K and 4.5 ± 0.5 log CFU/g for feed mill T. All samples tested negative for *E. coli* and *Salmonella* spp.

### 3.2. Effectiveness of SALTEC 514^TM^ to Prevent Salmonella Contamination in Fish Meal

The effects of SALTEC 514^TM^ on fish meal samples with *Salmonella* was investigated (Table 3). When a low dosage of SALTEC 514^TM^ was applied (0.5 kg/ton) to fish meal from feed mill T, the *Salmonella* recontamination was prevented for the longest study duration (97 days). The same low dosage of 0.5 kg/ton prevented the *Salmonella* recontamination in fish meal from feed mills SO and K for 37 days and feed mill SM for 31 days. In medium dosage applications of SALTEC 514^TM^ (1.0 kg/ton), the *Salmonella* recontamination was prevented for the longest study duration (97 days) in fish meal from feed mills SM and T. This dosage had also prevented *Salmonella* recontamination from taking place in fish meal from feed mills SO and K for 37 days. A high dosage application of SALTEC 514^TM^ (3.0 kg/ton) prevented the recontamination of *Salmonella* for the longest study time (97 days) in fish meal samples from feed mills SM, SO, and T (0% *Salmonella* occurrence), except for feed mill K where the *Salmonella* recontamination was prevented for 37 days (Table 3).

## 4. Discussion

Fish meal is recognized as a high-quality feed ingredient favorable for addition to the diet of most farm animals. It is of high energy content and an excellent source of protein, lipids, minerals, and vitamins [10,11]. The addition of fish meal to animal diets increases feed efficacy and growth performance by enhancing nutrient uptake, digestion, and absorption [12]. Fish meal can be produced from any type of wild-caught fish or small marine fish that contain a high percentage of bone and oil. It can also be produced from fish waste (by-product) of seafood processing factories. In addition, the sources of fish used for fish meal production depend on the location of the country and types of fish available for catching in each area. In Europe countries, fish including pout, capellin, sand eel, herring, and mackerel are mostly used for fish meal production while menhaden, anchovies, and pollock are commonly used in USA and South America. In Asian countries however, various species of fish caught from marine, freshwater, and brackish water are entirely used [13]. This may be due to a massive diversity of fish in the region.

High-quality fish meal should contain between 60% to 72% crude protein by weight [13]. Fish meal produced from Asian countries (India, Indonesia, Philippines, Thailand, and Vietnam) have an approximate range of crude protein from 40% to 75% [14]. Fish and its related products are one of the most perishable foods due to its peculiar chemical compositions. To maintain the freshness of fish meal during fish meal storage, the levels of chemical composition composed in fish meal should be monitored [15]. Total volatile basic nitrogen is one of the quality indices for fish meal to indicate bacterial spoilage in raw fish materials and finished fish meal. Ammonium, amine, trimethylamine (TMA), dimethylamine (DMA), and other volatile nitrogenous contents can be detected [16]. Total volatile basic nitrogen is produced during degradation of protein via enzymatic hydrolysis of spoilage bacteria, and autolysis reaction when fish meal is exposed to high temperatures. This affects fish meal quality and sensory aspects [15]. Total volatile basic nitrogen at 40 mg N/100 g meal is generally regarded as the limit of acceptability for premium quality fish meal [17,18]

Fish are natural hosts of various bacterial species found in the aquatic environment. Bacteria can be found on fish skin, gills, and the entire gastrointestinal tract due to the constant exposure and ingestion of contaminated feed and water [19]. Major strains of bacteria found in fish and its aquatic environment can be both pathogenic and non-pathogenic, belonging to several genera such as *Aeromonas*, *Acinetobacter*, *Alcaligenes*, *Edwardsiella*, *Enterobacter*, *Proteus*, *Providencia*, *Pseudomonas*, *Shewanella*, *Vibrio*, and especially *E. coli* [19,20,21,22,23].

The prevalence of *E. coli* in fish and its aquatic environments has been reported in several studies. Samples of freshwater fish originating from four Asian countries such as India, Myanmar, Thailand, and Vietnam showed a 27.2% total prevalence of extended-spectrum-β-lactamase (ESBL)-producing *E. coli* [24]. The highest prevalence (54.3%) was observed in milkfish samples from Vietnam, followed by catfish samples from Thailand (49.2%), and tilapia fish samples from India (37.3%) and Thailand (30.0%). However, no ESBL-producing *E. coli* was recovered from Rohu fish from Myanmar (0%). Similar to other reports, a high prevalence (36.0%) of *E. coli* compared to other bacteria was found in marine and freshwater fish samples derived from retail markets in Dar es Salaam, Tanzania [25]. The prevalence of *E. coli* has also been found in tropical seafood including finfish, shrimp, oysters, and clams. A high prevalence was observed in oysters (100%) and clams (78.0%) compared to other samples [26]. Contamination with *E. coli* is an important indicator for microorganisms in food, feed, and the environment [4,27,28].

*Salmonella* is not a normal flora of fish or aquaculture, they are widely found, and can survive and multiply in various natural conditions. Some species exist in animals without causing any disease symptoms. *Salmonella* has been detected in many types of feed ingredients, especially fish meal. The contamination can be initiated prior to or during feed mixing, or by cross-contamination during processing with inefficient hygiene practices, unsanitary equipment, and inadequate handling. Hence, feed is a potential source for *Salmonella* transmission to animals. Several studies have reported the prevalence of *Salmonella* from various fish sources, the main ingredient of fish meal production. From the previous report, the highest prevalence of *Salmonella* was observed from Nile perch (26%), followed by Nile tilapia (24%) and red snapper (12%) collected from the retail market in Dar es Salaam, Tanzania [25]. Out of 238 fish samples, 57 (24%) collected from Ouagadougou, Burkina Faso showed the positive result for *Salmonella* detection. The highest prevalence of *Salmonella* was related to the outbreak from serovars Bredeney and Muenster, where human activities could be linked to its contamination into products involving water (fish and vegetables) [29].

In the present study, no *E. coli* and *Salmonella* was detected in any samples collected from the feed mills. This may be due to sufficient hygiene practices, sanitary equipment, and proper handling at the feed mills. The process of cooking and drying at high temperatures in fish meal production could also be critical steps that reduced *E. coli* and *Salmonella* counts in the collected samples. However, the total viable count in all samples showed various levels of microbial load. This could be due to the survival and re-growth of microorganisms after processing with high temperatures and during storage. Cross-contamination by human sources may also be possible. In our further investigation, only *Salmonella* was selected as the target microorganism to test the effectiveness of SALTEC 514^TM^.

In the present study, we observed a variation in fish meal from different feed mills. The base formulation of the fish meal, including other components may contribute to a varying growth of *Salmonella* in our study. Awareness of the effect of bacteriostatic agents on the recovery of organisms is important when determining product efficacy [30]. Some masking effects were observed when a high count of *Salmonella* was present in the feed [31]. Masking can be caused by the organic acid lowering the pH of the culture media, thus causing injury or death to *Salmonella* during culture, with the effect of this varying between serovars and feed matrices [32,33].

SALTEC 514^TM^ is a synergistic combination of organic acids in its free form with surfactants, containing up to 85% (*v*/*v*) total acids. The major effective chemical ingredients for SALTEC 514^TM^ formulation are formic acid and propionic acid. These organic acids have been widely applied in food and feed materials as an antimicrobial agent against a wide range of microorganisms, especially foodborne pathogens. The mode of action of organic acids is thought to be pH dependent. They influence the redox reductions in NADPH formation, which is particularly required as an energy source in microorganisms. Organic acids can penetrate freely across the cell membrane of bacteria and release hydrogen ions (H+) and cause the reduction of the internal pH of a bacterial cell. Organic acids also destabilize the structure of several proteins such as enzymes and cell wall components of microorganisms leading to protein malfunction and cell wall destruction [34,35]. Organic acids are widely applied in food and feed ingredients to inhibit a wide range of microorganisms. For example, formic acid and propionic acid demonstrated the highest inhibition of 2.5 log units of *Salmonella* in pelleted and compound mash feed after 5 days of exposure [36]. In another study, different concentrations of acetic acid, lactic acid, propionic acid, and formic acid was applied to meat contaminated with *E. coli* and *Staphylococcus aureus* via spray wash. It was found that the mean log reduction ranged from 0.89 to 1.84 log units of *E. coli* and 1.15 to 3.16 log units of *S. aureus* after 12 days of storage in a refrigerator [37]. The results of the present study demonstrated that treatment of fish meal with organic acids of SALTEC 514^TM^ controlled and prevented *Salmonella* recontamination. In addition, the use of organic acids in the formulation of SALTEC 514^TM^ presented a safe option for feed production.

## 5. Conclusions

In the present study, SALTEC 514^TM^, a synergistic combination of organic acids in its free form with surfactants, containing up to 85% (*v*/*v*) total acids, showed superior antimicrobial activity against *Salmonella* in fish meal. Fish meal samples recontaminated with *Salmonella* and stored at an ambient temperature may remain *Salmonella* free for up to 97 days when SALTEC 514^TM^ was applied at a low dosage (0.5 kg/ton). The inhibitory effect of the antimicrobial agent is derived from its organic acid formulation. The application of SALTEC 514^TM^ in fish meal and/or other feed ingredients may prove to be a safe alternative to reduce the microbial load, especially of foodborne-related microorganisms, to contribute to feed and food safety.

## Figures and Tables

**Table 1 animals-12-03367-t001:** Protein and total volatile basic nitrogen contents of fish meal samples from feed mills.

Fish Meal	Protein Content (g/100 g) ^1^	Total Volatile Basic Nitrogen (mg/100 g) ^1^
SM	53.2 ± 3.1	87.1 ± 5.7
SO	63.5 ± 4.4	73.8 ± 4.5
T	67.5 ± 2.3	100.3 ± 9.2
K	65.6 ± 8.1	100.4 ± 11.2

^1^ All values were reported as mean ± standard deviation (SD) (*n* = 3). SM, SO, T, and K were assigned name of fish meal samples.

**Table 2 animals-12-03367-t002:** Total plate count and occurrence of *Salmonella* and *E. coli* in fish meal samples from different feed mills.

Fish Meal	Total Plate Count (Log Colony Forming Unit/g) ^1^	*Escherichia coli*	*Salmonella* spp.
SM	2.3 ± 0.2	ND	ND
SO	2.0 ± 0.3	ND	ND
T	4.5 ± 0.5	ND	ND
K	4.2 ± 0.4	ND	ND

^1^ All values were reported as mean ± standard deviation (SD) (*n* = 3). ND: not detected.

**Table 3 animals-12-03367-t003:** Inhibitory effects of SALTEC 514^TM^ against *Salmonella* recontamination in fish meal.

Fish Meal	Dosage	*Salmonella* Counts (Log Colony Forming Unit/g) at Each Storage Time ^1^						
		Before *Salmonella* Challenge			Challenge at Day 30		Challenge at Day 90	
		Day 0	Day 1	Day 7	Day 31	Day 37	Day 91	Day 97
SM	Control	2.9 ± 0.1 ^aA^	3.0 ± 0.3 ^bA^	4.0 ± 0.1 ^cA^	2.9 ± 0.2 ^aA^	4.1 ± 0.1 ^cA^	3.7 ± 0.1 ^cA^	3.8 ± 0.3 ^cA^
	Low	3.1 ± 0.3 ^aA^	0.0 ± 0.0 ^cB^	0.0 ± 0.0 ^cB^	0.0 ± 0.0 ^cB^	2.5 ± 0.1 ^bB^	2.5 ± 0.2 ^bB^	3.4 ± 0.4 ^aA^
	Medium	3.0 ± 0.1 ^aA^	0.0 ± 0.0 ^bB^	0.0 ± 0.0 ^bB^	0.0 ± 0.0 ^bB^	0.0 ± 0.0 ^bC^	0.0 ± 0.0 ^bC^	0.0 ± 0.0 ^bB^
	High	3.1 ± 0.2 ^aA^	0.0 ± 0.0 ^bB^	0.0 ± 0.0 ^bB^	0.0 ± 0.0 ^bB^	0.0 ± 0.0 ^bC^	0.0 ± 0.0 ^bC^	0.0 ± 0.0 ^bB^
SO	Control	2.8 ± 0.3 ^aA^	3.6 ± 0.2 ^bA^	3.8 ± 0.1 ^bcA^	3.6 ± 0.2 ^bA^	3.6 ± 0.2 ^bA^	4.2 ± 0.1 ^cA^	3.2 ± 0.2 ^bB^
	Low	3.1 ± 0.2 ^bA^	0.0 ± 0.0 ^cB^	0.0 ± 0.0 ^cB^	0.0 ± 0.0 ^cB^	0.0 ± 0.0 ^Cb^	3.8 ± 0.2 ^aA^	3.8 ± 0.1 ^aA^
	Medium	2.8 ± 0.2 ^bA^	0.0 ± 0.0 ^cB^	0.0 ± 0.0 ^cB^	0.0 ± 0.0 ^cB^	0.0 ± 0.0 ^cB^	3.7 ± 0.1 ^aA^	3.7 ± 0.4 ^aA^
	High	2.9 ± 0.4 ^aA^	0.0 ± 0.0 ^bB^	0.0 ± 0.0 ^bB^	0.0 ± 0.0 ^bB^	0.0 ± 0.0 ^bB^	0.0 ± 0.0 ^bB^	0.0 ± 0.0 ^bC^
T	Control	2.7 ± 0.4 ^aA^	3.1 ± 0.2 ^bA^	3.8 ± 0.3 ^cA^	2.5 ± 0.1 ^aA^	2.5 ± 0.1 ^aA^	2.8 ± 0.3 ^abA^	2.8 ± 0.2 ^abA^
	Low	3.1 ± 0.1 ^aA^	0.0 ± 0.0 ^bB^	0.0 ± 0.0 ^bB^	0.0 ± 0.0 ^bB^	0.0 ± 0.0 ^bB^	0.0 ± 0.0 ^bB^	0.0 ± 0.0 ^bB^
	Medium	3.3 ± 0.4 ^aA^	0.0 ± 0.0 ^bB^	0.0 ± 0.0 ^bB^	0.0 ± 0.0 ^bB^	0.0 ± 0.0 ^bB^	0.0 ± 0.0 ^bB^	0.0 ± 0.0 ^bB^
	High	3.1 ± 0.2 ^aA^	0.0 ± 0.0 ^bB^	0.0 ± 0.0 ^bB^	0.0 ± 0.0 ^bB^	0.0 ± 0.0 ^bB^	0.0 ± 0.0 ^bB^	0.0 ± 0.0 ^bB^
K	Control	2.8 ± 0.3 ^aA^	3.4 ± 0.1 ^abA^	4.2 ± 0.2 ^bA^	4.7 ± 0.1 ^bcA^	4.6 ± 0.2 ^bcA^	3.3 ± 0.1 ^abA^	5.0 ± 0.4 ^cA^
	Low	2.9 ± 0.1 ^aA^	0.0 ± 0.0 ^bB^	0.0 ± 0.0 ^bB^	0.0 ± 0.0 ^bB^	0.0 ± 0.0 ^bB^	3.2 ± 0.2 ^aA^	4.1 ± 0.1 ^aB^
	Medium	3.1 ± 0.2 ^bA^	0.0 ± 0.0 ^bB^	0.0 ± 0.0 ^bB^	0.0 ± 0.0 ^bB^	0.0 ± 0.0 ^bB^	3.1 ± 0.2 ^bA^	3.7 ± 0.2 ^aB^
	High	3.0 ± 0.2 ^aA^	0.0 ± 0.0 ^cB^	0.0 ± 0.0 ^cB^	0.0 ± 0.0 ^cB^	0.0 ± 0.0 ^cB^	2.9 ± 0.2 ^aA^	2.3 ± 0.1 ^bC^

^1^ All values were reported as mean ± standard deviation (*n* = 3). The lowercase letters (^a, b, c^) for *Salmonella* count at each storage time (same row) and the uppercase letters (^A, B, C^) for *Salmonella* count in each dose with the same fish meal (same column), which connected by the different letters are significantly different (*p* < 0.05).

## Data Availability

The data presented in this study are available on request from the corresponding author.

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
