# Peer review of "Effectiveness of the Organic Acid-Based Antimicrobial Agent to Prevent Bacterial Contamination in Fish Meal"

_animals, 2022, doi:10.3390/ani12233367_

Round 1

Reviewer 1 Report

The study by Pelyuntha et al. investigated the efficacy of the organic acid-based antimicrobial agent SALTEC-514TM against Salmonella in preventing recontamination in fish meals. From the data presented by the author, there is no doubt about the effectiveness of SALTEC-514TM premix for preventing Salmonella spp and controlling microbial contamination in fish meals.  However, there are some concerns about the safety and health implication of using this compound as a food additive.  Is this compound safe for humans consuming fish? How about the doses? How many times does this compound need to add to food? What is the chance for bacteria to get resistance toward this compound? Is it licensed as premix to control bacteria in animal food, and in which countries has it been used? Do you have data regarding residues in fish? In my opinion, the author must consider discussing all these issues in the discussion section

Specific comment:

Line 101: AOAO, please add the full name of this abbreviation

Line 165: Please put the bacteria name in italic

Line 244, 247, 248, and 252: Please remove the extra space

Author Response

#Response to Reviewer 1

The study by Pelyuntha et al. investigated the efficacy of the organic acid-based antimicrobial agent SALTEC-514TM against Salmonella in preventing recontamination in fish meals. From the data presented by the author, there is no doubt about the effectiveness of SALTEC-514TM premix for preventing Salmonella spp and controlling microbial contamination in fish meals.  However, there are some concerns about the safety and health implication of using this compound as a food additive. Is this compound safe for humans consuming fish? How about the doses? How many times does this compound need to add to food? What is the chance for bacteria to get resistance toward this compound? Is it licensed as premix to control bacteria in animal food, and in which countries has it been used? Do you have data regarding residues in fish? In my opinion, the author must consider discussing all these issues in the discussion section

Answer: SALTEC-514TM is the formulation of two organic acids, formic acid and propionic acid combined with surfactant. This product is used as feed additive for preventing the microbial contamination in feed and feed raw ingredient. Formic acid and propionic acid are general recognized as safe (GRAS) by U.S. FDA and other food authorities worldwide.

Propionic acid and its salt are commonly used as antimicrobial agent for foods, feed, human & animal drugs, cosmetics, and related products. There was no limited acceptable daily intake (ADI) for propionic acid for human and animal consumption.

As same as formic acid, this acid and its salt forms are commonly used as animal feed additive and/or mixing with drinking water of animals. It is permitted for use as a component of synthetic flavoring substances and adjuvant permitted for direct addition to food for human consumption. Formic acid can be found as a normal component of human blood and tissue. However, acceptable daily intake (ADI) of formic acid is 0-3 mg/kg for human consumption (by JECFA) while there was no limitation used for animal feed additive (by EFSA).

References:

  • https://www.accessdata.fda.gov/scripts/cdrh/cfdocs/cfcfr/CFRSearch.cfm?fr=582.3081
  • https://www.accessdata.fda.gov/scripts/cdrh/cfdocs/cfcfr/CFRSearch.cfm?fr=582.3081
  • https://www.fda.gov/files/food/published/GRAS-Notice-000668---Sodium-formate.pdf
  • https://www.federalregister.gov/documents/2017/11/13/2017-24366/food-additives-permitted-in-feed-and-drinking-water-of-animals-ammonium-formate-and-formic-acid
  • https://www.accessdata.fda.gov/scripts/cdrh/cfdocs/cfcfr/CFRSearch.cfm?fr=186.1316&SearchTerm=formic%20acid
  • https://www.accessdata.fda.gov/scripts/cdrh/cfdocs/cfcfr/CFRSearch.cfm?fr=573.480
  • https://inchem.org/documents/jecfa/jeceval/jec_2015.htm

Specific comment:

Line 101: AOAO, please add the full name of this abbreviation

Answer: Full name of AOAC was added (Line 113-114).

Line 165: Please put the bacteria name in italic

Answer: Bacterial name was italicized.

Line 244, 247, 248, and 252: Please remove the extra space

Answer: Extra spaces were removed.

Reviewer 2 Report

This study explored the efficacy of the organic acid-based antimicrobial agent against Salmonella to prevent recontamination fish meal. The topic of this study is practical important, and the results can be applied in feed industry. The general design of this study is sound, and the authors have obtained a series good result. It is suggested that the author revise the part of statistical methods to clarify the specific methods.

Abstract

Overall, the design of this study was not clear in the abstract, please add the design of this study in the absreact.

Line 25: “the organic acid-based antimicrobial agent SALTEC-514TM”,is it commercial product or raw materials?

Introduction

Lines 79-84: It is suggested to supplement the effective chemical composition of "SALTEC-514TM". Please also add the objective of this study in the end of the introduction.

Materials and method

Lines 140-144: there were three parameters (fishmeal source, dose, time) in this study, author should clearly indicate their analysis method in this section.

Results

Tables: It is recommended to add the full name of each abbreviation at the bottom of the table.

Author Response

#Response to Reviewer 2

This study explored the efficacy of the organic acid-based antimicrobial agent against Salmonella to prevent recontamination fish meal. The topic of this study is practical important, and the results can be applied in feed industry. The general design of this study is sound, and the authors have obtained a series good result. It is suggested that the author revise the part of statistical methods to clarify the specific methods.

Abstract

Overall, the design of this study was not clear in the abstract, please add the design of this study in the abstract.

Answers: Some points of design of this study were added for clarification according to the reviewer’s suggestion (Line 24-26, 32-36).

Line 25: “the organic acid-based antimicrobial agent SALTEC-514TM”, is it commercial product or raw materials?

Answer: SALTEC-514TM is a commercial product used as a preservative for raw materials.

References for product: http://www.linqtec.com/en/saltec-514-3/

Introduction

Lines 79-84: It is suggested to supplement the effective chemical composition of "SALTEC-514TM". Please also add the objective of this study in the end of the introduction.

Answer: The effective chemical compositions used for SALTEC-514TM formulation (formic acid and propionic acid with surfactant) was added. The objective was added to the manuscript according to the reviewer’s comment (Line 81, 84-87).

Materials and method

Lines 140-144: there were three parameters (fishmeal source, dose, time) in this study, author should clearly indicate their analysis method in this section.

Answer: Thank you for your comment. The statistical analysis was revised according to the reviewer’s suggestion. We only considered two parameters (dose and storage time) within the same fishmeal sources (Table 3).

Results

Tables: It is recommended to add the full name of each abbreviation at the bottom of the table.

Answer: An abbreviation was added to the manuscript according to the reviewer’s comment (Table 2).

Reviewer 3 Report

The research question is important. The contamination of raw ingredients with Salmonella in feed production is an important food safety issue. It is necessary to avoid the spread of pathogens into food animals. The study investigated the use of a liquid antimicrobial agent containing organic acids (SALTEC-514TM) to reduce the risk of microbial contamination in feed ingredients containing fish meal samples.

The study used six fish meal samples. The experimental methods included the determination of TVBN contents as well as bacteriological analyses to determine total bacterial counts and the occurrence of Salmonella. It was also included the experiment with the Salmonella artificial contamination. The Results are organized in sections to describe the protein and TVBN contents as well as the bacteriological results in the fish meals and the contaminated ones over time. 

In general, the manuscript is relatively well organized and the whole message is clear. However I think it is not ready for publication. It has many flaws that prevent the acceptance right now. I am highlighting some concerns below:

1)    Title: it is not Ok since the fish meals were not “recontaminated”. I think something like "Effectiveness of an organic acid-based antimicrobial agent to prevent bacterial contamination in fishmeal" is more appropriate.

2)    Samples: why do the authors describe six fish meals if they used only four in the experiments? I suggest removing the two samples (P and M)  from  the study.

3)    Samples: the protein and TVBN contents of all samples must be described in the methods section and not in the Results. So the authors should remove the first Results section that describes the fish meals samples characteristics. 

4)    Methods: it is described that total plate count and detection of Escherichia coli were performed. In addition, the authors described that “Pink colonies were further observed and enumerated”. What does it mean the occurrence of “pink colonies”? Please explain.

5)    Results: The section 3.1 should be removed (see 3 above).

6)    Results: The titles of the sections 3.2 and 3.3 should be reviewed. A) It was not demonstrated the “prevalence” of E. coli and Salmonella in fish meals (“occurrence” would be more appropriate). B) Fish meals were not “recontaminated”. Please also use shorter section titles.

7)    Results: I suggest to present the Results in bar graphs instead of tables.

Discussion: I suggest the authors to focus the discussion on the contamination of fish meals. I think it is not necessary to compare protein and TVBN contents of the fish meal samples. So two paragraphs must be removed. 

Author Response

#Response to Reviewer 3

The research question is important. The contamination of raw ingredients with Salmonella in feed production is an important food safety issue. It is necessary to avoid the spread of pathogens into food animals. The study investigated the use of a liquid antimicrobial agent containing organic acids (SALTEC-514TM) to reduce the risk of microbial contamination in feed ingredients containing fish meal samples.

The study used six fish meal samples. The experimental methods included the determination of TVBN contents as well as bacteriological analyses to determine total bacterial counts and the occurrence of Salmonella. It was also included the experiment with the Salmonella artificial contamination. The Results are organized in sections to describe the protein and TVBN contents as well as the bacteriological results in the fish meals and the contaminated ones over time.

In general, the manuscript is relatively well organized and the whole message is clear. However I think it is not ready for publication. It has many flaws that prevent the acceptance right now. I am highlighting some concerns below:

1)    Title: it is not Ok since the fish meals were not “recontaminated”. I think something like "Effectiveness of an organic acid-based antimicrobial agent to prevent bacterial contamination in fishmeal" is more appropriate.

Answer: Title was changed according to the reviewer’s suggestion.

2)    Samples: why do the authors describe six fish meals if they used only four in the experiments? I suggest removing the two samples (P and M) from the study.

Answer: Samples P and M were removed from this study according to the reviewer’s suggestion.

3)    Samples: the protein and TVBN contents of all samples must be described in the methods section and not in the Results. So, the authors should remove the first Results section that describes the fish meals samples characteristics.

Answer: Actually, the protein and TVBN contents of fish meal were investigated in this study using a commercial service. These parameters can be used for indirect determination of fish meal quality, especially microbiological attribute that bacteria involved in contamination could degrade the nutrition (protein) present in fish meal into TVBN (as described in Discussion section). However, the fish meal characteristics (protein and TVBN content) was revised according to the reviewer’ suggestion (move to Section 2.2).

4)    Methods: it is described that total plate count and detection of Escherichia coli were performed. In addition, the authors described that “Pink colonies were further observed and enumerated”. What does it mean the occurrence of “pink colonies”? Please explain.

Answer: We observed the presence of pink colonies of E. coli presented in the chromogenic agar plate. This modified method from bioMérieux company can use for both qualitative and quantitative analysis of Escherichia coli in sample. Hence, we can observe the occurrence of E. coli in the samples along with the number by counting method. However, the sentence was revised according to the reviewer’s comment.

5)    Results: The section 3.1 should be removed (see 3 above).

Answer: Section 3.1 (Protein and TVBN) was removed according to the reviewer’s comment.

6)    Results: The titles of the sections 3.2 and 3.3 should be reviewed. A) It was not demonstrated the “prevalence” of E. coli and Salmonella in fish meals (“occurrence” would be more appropriate). B) Fish meals were not “recontaminated”. Please also use shorter section titles.

Answer:

(A) “Prevalence’ was changed to “Occurrence” according to the reviewer’s suggestion.

(B) Section title was changed according to the reviewer’s suggestion.

7)    Results: I suggest to present the Results in bar graphs instead of tables.

Answer: Thank you for your suggestion. We would like to keep the result as Table. It looks better than the bar graph due to the excessive presentation as Figures (We think they will make a confusion) and the overlap of result (0 log CFU/g) in legend at each dose at the same time.

Discussion: I suggest the authors to focus the discussion on the contamination of fish meals. I think it is not necessary to compare protein and TVBN contents of the fish meal samples. So two paragraphs must be removed.

Answer: We decided to remove the comparison of protein and TVBN result with other previous reports according to the reviewer’s suggestion. However, we would like to keep some points of this discussion because of these attributes (protein and TVBN) are important to determine the bacterial contamination in high-protein feed ingredients. Is it possible?

Round 2

Reviewer 3 Report

The authors made the main changes that were recommended. The article can be accepted in its current version.